# Biomarkers of Kidney Disease in Horses: A Review of the Current Literature

**DOI:** 10.3390/ani12192678

**Published:** 2022-10-05

**Authors:** Gaby van Galen, Emil Olsen, Natalia Siwinska

**Affiliations:** 1Sydney School of Veterinary Science, University of Sydney, Sydney, NSW 2006, Australia; 2Goulburn Valley Equine Hospital, Congupna, VIC 3633, Australia; 3Veterinary Teaching Hospital (Universitetsdjursjukhuset, UDS), Swedish Veterinary Agricultural University (SLU), 750 07 Uppsala, Sweden; 4Department of Internal Medicine, Clinic of Diseases of Horses, Dogs and Cats, Faculty of Veterinary Medicine, University of Environmental and Life Sciences Wroclaw, 50-375 Wroclaw, Poland

**Keywords:** renal, equine, SDMA, NGAL, cystatin C, podocin, NAG, MMP, validation, reference range, assay

## Abstract

**Simple Summary:**

The one marker that we routinely use to diagnose kidney disease in horses (creatinine) is not an ideal marker. Having additional biomarkers in blood and urine that can be measured could be beneficial. This review assesses available literature on new equine kidney biomarkers, and the authors suggest recommendations for clinical use in equine medicine. The markers discussed here are SDMA, cystatin C, podocin, NGAL, MMP-9 and NAG. NGAL, cystatin C and podocin show potential as biomarkers for kidney disease (including detecting kidney injury earlier than creatinine). SDMA may have some potential as a biomarker for equine kidney disease, but there is a lack of evidence for SDMA being advantageous over creatinine. For all biomarkers discussed here, the available scientific information is limited or too scarce to support clinical use, and only SDMA can be measured for clinical purposes. In conclusion, there are multiple new biomarkers with the potential to assess kidney function better than creatinine. However, there are only few studies available, and more data is needed before these biomarkers can be applied and recommended in our daily practice.

**Abstract:**

Creatinine only allows detection of kidney disease when 60 to 75% of the glomerular function is lost and is therefore not an ideal marker of disease. Additional biomarkers could be beneficial to assess kidney function and disease. The objectives are to describe new equine kidney biomarkers. This systematic review assesses the available literature, including the validation process and reference values, following which the authors suggest recommendations for clinical use. SDMA may have some potential as equine kidney biomarker, but there is currently a lack of evidence that SDMA offers any advantage compared to creatinine in detecting Acute Kidney Injury (AKI). Cystatin C and podocin show potential as biomarkers for kidney disease (including detecting AKI earlier than creatinine) and should be studied further. NGAL has potential as a biomarker of kidney disease (including detecting AKI earlier than creatinine), and potential as an inflammatory marker. Literature on MMP-9 does not allow for conclusive statements about its potential as a biomarker for kidney disease. The future may show that NAG has potential. For all biomarkers, at this stage, available scientific information is limited or too scarce to support clinical use, and only SDMA can be measured for clinical purposes. In conclusion, there are multiple new biomarkers with the potential to diagnose kidney problems. However, there are only a few studies available and more data is needed before these biomarkers can be applied and recommended in our daily practice.

## 1. Introduction

The use of terminology around kidney disease in horses has changed in recent years. Therefore, the authors will start by briefly introducing and defining the terms that are currently relevant and that will be used throughout this manuscript: Acute Kidney Injury (AKI), Chronic Kidney Disease (CKD) structural versus functional disease, prerenal/renal/postrenal azotemia, and biomarker.

Acute Kidney Injury (AKI) represents a continuum of acute kidney injury from mild, clinically inapparent nephron loss to severe acute kidney failure. In equine medicine, there is currently a lack of consensus on the definition of AKI. In human medicine, AKI is defined as an increase in the serum creatinine by 0.3 mg/dL (26.5 micromol/L) or more within 48 h, or an increase in serum creatinine over 1.5 × the baseline or more within 7 days, or oliguria <0.5 mL/kg/h for 6 h [1]. Applying these human criteria in an equine referral setting in England, AKI occurred with a prevalence of 14.8% [2].

Chronic Kidney Disease (CKD) is a chronic, irreversible, progressive disease of the kidneys, with a duration greater than 3 months. In human and small animal literature, the term Chronic Renal Failure has, over the past 20 years, been replaced with Chronic Kidney Disease (CKD), because the word “failure” reflects an end-stage where the kidney function is impaired to a point where the patient cannot survive without kidney transplantation or dialysis. Kidney disease can be chronic without being end-stage, or even without clinical functional loss. Therefore, the term Chronic Kidney Disease is believed to be more appropriate, and has recently also been applied to equine medicine by Olsen and van Galen [3]. For horses, CKD can be defined as a creatinine >180 micromol/L (or >2.0 mg/dL) lasting longer than 3 months [3].

Both AKI and CKD can be graded according to severity, and an example of how that can be done for horses with CKD has recently been suggested for the first time by Olsen and van Galen [3]. Apart from extrapolations from human and small animal scorings in research papers, there has not yet been a clear suggestion or consensus for a scoring system for AKI in horses.

Kidney disease is typically defined as functional or structural, where both can result in a reduction in the glomerular filtration rate (GFR). Structural kidney disease can be palpated or imaged whereas functional kidney disease often occurs with none or few palpatory or ultrasound changes. Structural disease (or damage) usually precedes alterations in function and can occur without loss of function or with loss of function based on the extent of the disease.

Azotemia is an umbrella term for abnormally increased concentrations of urea, creatinine, and other nonprotein nitrogenous substances in the blood. Measuring creatinine and urea are considered clinically the easiest way to estimate GFR. However, the presence of azotemia is not synonymous with loss of kidney function, and it is possible to have azotemia without kidney disease. Prerenal (decreased perfusion, e.g., dehydration, hypovolemic shock, cardiac failure), renal (decreased GFR caused by renal injury), or postrenal causes of azotemia (failure of excretion of nitrogenous waste such as obstruction of the urinary tract or rupture of the urinary bladder) should be differentiated. Prerenal causes for azotemia are often mentioned in equine literature, however, following human literature, and in the authors’ opinions, pure prerenal causes are rare. Most prerenal causes are closely linked with or intertwined with inflammatory changes and/or toxic insults. For example, a horse presented with acute colic and cardiovascular shock will also have systemic inflammatory response syndrome (SIRS) or endotoxemia and, at its most vulnerable moment, receive nephrotoxic drugs (flunixin and aminoglycosides such as gentamicin are the most common nephrotoxic drugs administered to horses undergoing exploratory laparotomy). Such a high-risk patient with SIRS and hypovolemia, therefore, has compromised perfusion as well as inflammatory and toxic insults to the kidney. Furthermore, there is a continuum between prerenal and renal azotemia, where processes that affect perfusion, dependent on severity and duration, will eventually lead to intrinsic kidney damage. If the azotemia is renal in nature, it should be determined if the changes are acute, acute-on-chronic, or chronic based on duration. AKI and CKD are also possible without azotemia, for example in unilateral disease where a healthy kidney can compensate for the partial loss of function of affected kidney, and/or damage that is impacting a small amount of nephrons (less than the threshold for creatinine concentrations to increase).

Glomerular filtration rate (GFR) is the reference standard to assess kidney function. Creatinine has traditionally been used as an indicator of GFR as it is produced in the muscle from creatine and filtered in the glomeruli. When GFR decreases, serum creatinine increases, which indicates loss of kidney function. Unfortunately, by the time creatinine is increased above reference range, approximately 60–75% of the functional nephrons are lost [4], and therapeutic interventions are therefore often late. Serum creatinine varies markedly between individuals and with age and breed; it is higher in some normal Quarter Horses [5] with greater muscle mass and lower in cachexic animals with subnormal muscle mass. However, individual variation is low, which allows to follow changes in kidney function over time. In early kidney disease, a small increase in serum creatinine concentration indicates a large decrease in GFR, whereas in advanced kidney disease a large increase indicates smaller changes in GFR. Therefore, small increases in creatinine are often of limited clinical relevance in patients with CKD, whereas they are very important in AKI. Because of all these reasons, creatinine is far from an ideal marker to assess such a highly critical organ. Additional biomarkers to assess patients with kidney disease or monitor patients at risk for kidney disease could be beneficial, especially if they can identify kidney disease earlier than creatinine. Furthermore, creatinine is a functional marker, and including markers of structural damage would allow for a more complete assessment of the kidneys, similar to hepatic assessments. Other traditional kidney measures include blood urea nitrogen, urinary GGT, urinary protein/creatinine ratio, urinary specific gravity (USG), and fractional excretion of sodium and other electrolytes.

As defined in 1998 by the National Institutes of Health Biomarkers Definitions Working Group, a biological marker, commonly known as a biomarker, is “*a characteristic that is objectively measured and evaluated as an indicator of normal biological processes, pathogenic processes, or pharmacologic responses to a therapeutic intervention*” [6]. The World Health Organization (WHO) in cooperation with the United Nations has extended the existing definition as “*any substance, structure or process that can be measured in the body or its products and influence or predict the incidence of outcome or disease*” [7]. In medicine, there are 7 categories of biomarkers: risk, diagnostic, monitoring, predictive, prognostic, pharmacodynamic and safety. Most often, however, a single biomarker belongs to many categories at the same time [8]. An ideal biomarker of kidney disease has the possibility to: early detect kidney injury, even before the increase in serum creatinine concentration; distinguish acute from chronic processes; distinguish injury location, ie tubular injury from glomerulonephritis or interstitial nephritis; monitoring the response to treatment; predicting mortality and long-term prognosis. Additional criteria for biomarkers of kidney disease are: ease of measuring its presence or concentration in blood or urine; stability in body fluids; specificity—production by the kidneys, preferably at the site of injury [9].

In human medicine multiple biomarkers for kidney disease have been discovered and compared to the reference standard, allowing a more detailed diagnostic workup. Progress in biomarker use in human medicine is often extrapolated to veterinary medicine, however, such extrapolation can be flawed by many hurdles related to dissimilarities between species and pathophysiology. Before new biomarkers can reliably be used in a different species and can be considered useful in a clinical setting in a different species, there are 4 phases of a complete assay validation for diagnostic purposes, as described by Kelgaard-Hansen and Jacobsen [10].

i.Phase I: Analytical Performance, assessment of analytical and practicability characteristics. How well does the test function from an analytical point of view, ie intra- and inter-assay variations, linearity under dilution, and other factors.ii.Phase II: Overlap Performance, assessment of value in health and disease. Are there differences in concentration of the biomarker between different health groups (the diagnosis is pre-determined and the biomarker is used in retrospect)?iii.Phase III: Clinical Performance, evaluation of diagnostic sensitivity and specificity in clinical settings of interest. How well does the biomarker allow for reaching a diagnosis of a health group compared to a reference standard?iv.Phase IV: Outcome and Usefulness Performance, assessment of whether the individual patient (or community) gain advantage from the test. Is it an advantage for patient management (faster, cheaper, more precise, better outcomes,…) to integrate the biomarker in the diagnostic approach? This phase is rarely performed in veterinary medicine.

Application of new diagnostic biomarkers in veterinary medicine seldom awaits a full 4-phase validation, and unfortunately often skips analytical validation. Consequently, it is important to take this 4 phase-process into consideration when assessing the literature of new biomarkers of kidney disease and evaluating clinical applicability of new biomarkers. In addition to the validation process, reference ranges need to be determined, for which guidelines are available from the veterinary pathology college [11]. Nonparametric methods with calculations of 90% confidence intervals of the upper and lower reference limit are recommended when at least 120 samples are available [11]. Reference ranges can also be determined on smaller sample sizes (minimum sample size of 20) with different statistical methods [11]. Extrarenal factors such as breed, age, sex, body weight can influence reference ranges, and also need to be investigated before a biomarker can reliably be applied in a clinical setting.

Therefore, the objectives of this systematic review are to describe new biomarkers of kidney disease in equine medicine, the current physiological knowledge that is available on them, the status of their validation process including determination of reference values, and if, at this point in time, they can be recommended for clinical use.

## 2. Materials and Methods

This systematic review has been executed following PRISMA guidelines (where applicable).

### 2.1. Search for Literature to Be Included in this Review

Eligibility criteria for studies to be included in this systematic review were: (1) including original data on a new biomarker for kidney disease, (2) involving equine patients or equine samples (all ages, sex and breeds included), (3) peer-reviewed published studies. Exclusion criteria were: (1) studies that only include data on traditional measures of kidney function, (2) studies on other species, (3) review papers or opinion pieces. Data from non-peer reviewed published studies, such as those presented at international conferences and published as proceeding notes, were also not included in the search nor included in this systematic review. To make the readers aware of ongoing research efforts on these biomarkers, a list of published proceeding notes was provided in the section on the specific biomarker, without discussing the data.

A literature search was performed through a Pubmed search, with the following search words: renal biomarker combined with horse or equine. In addition, Pubmed was searched with the names of renal biomarkers known to the authors in combination with horse or equine: SDMA, NGAL, cystatin C, podocin. The date of the last Pubmed search was the 16th of August 2022. The Pubmed search and selection of studies to be included was conducted by the first author, and was double checked by the other authors.

### 2.2. Data Retrieved from Literature and Data Analysis

A list was made with all biomarkers for kidney disease in horses identified in the literature. Studies were grouped according to the biomarker that was described. A study describing more than one biomarker was categorised in all relevant biomarker groups, and therefore can appear multiple times in this review. For each study, the following was reported: study characteristics (including assay used, number of horses included and horses with AKI/CKD, sample type used) and study results (comparison to traditional tools for assessment of kidney function, investigation in extrarenal factors affecting reference ranges, association to clinical disease state, phase of biomarker assay validation, reporting of concentrations of healthy horses/reference ranges), and limitations of the study.

For all biomarkers the variables mentioned in Table 1 were described from individual studies, and data from the individual studies was summarised.

## 3. Results

### 3.1. Results of Literature Search

Through the Pubmed search 18 studies were identified to be included in this systematic review (Figure 1).

The biomarkers identified through the literature search include: symmetric dimethylarginine (SDMA; 9 studies), neutrophil gelatinase-associated lipocalin (NGAL; 5 studies), cystatin C (3 studies), podocin (1 study), MMP (1 study), *N*-acetyl-beta-D-glucosaminidase (NAG; 1 study).

### 3.2. Current Knowledge and Level of Evidence 

#### 3.2.1. Markers of Kidney Function (Estimate of GFR)

##### Symmetric Dimethylarginine (SDMA)

In humans and in small animals, SDMA is recognized as a useful, albeit highly debated, biomarker for kidney function. It has been found to have good correlations with GFR and creatinine and with increases in concentration seen earlier than creatinine [12].

Nine studies in total report on SDMA in horses (Table 2). Several of these include data on analytical validation of the test that was used [13,14,15,16]. All of the used assays have undergone analytical validation (phase I of the validation process). In the different studies, different assays were used, amongst which a commercially available assay that was developed by IDEXX and is available for clinical use. Freezing has no negative impact on SDMA analysis [17].

Several of these studies include overlap performance data (phase II). These showed different serum concentrations of SDMA between (1) healthy horses versus horses with AKI [18,19], (2) horses with different levels of dehydration [19], and (3) endurance horses before versus after the race, with post-race SDMA concentrations being higher with longer racing distance [13]. There was no difference in serum SDMA concentrations between healthy horses versus horses with cardiac disease [14], or between survivors and non-survivors with different degrees of dehydration (lower SDMA concentrations were found in survivors than in non-survivors, however this was not statistically different) [19]. SDMA concentrations have only been reported on very few equine AKI cases (37 cases divided over 4 studies) [15,16,18,19], and so far on no confirmed equine CKD cases. One study did not identify different SDMA concentrations between healthy horses and horses at risk for AKI (and therefore potential early AKI), whereas some traditional markers (urinary protein and urine protein to creatinine ratio (UP/UC) and urinary GGT/creatinine ratio) differed [18]. Two other studies reported 2 and 7 cases with elevated SDMA but normal creatinine; these cases were interpreted as potential yet unconfirmed subclinical AKI or CKD cases [15,19]. Therefore, similar as to small animals, there is a possibility that SDMA is beneficial to detect early equine AKI cases, when creatinine is still within reference range, yet evidence is currently lacking. In cats and dogs, SDMA increases in response to a 40% decrease in GFR [12]. It is currently not known for horses how much loss of function is needed before SDMA increases, and if it is an earlier marker for AKI than creatinine.

There are no equine studies that report on optimal cut off values of SDMA and its sensitivity and specificity as a diagnostic tool for AKI or CKD in horses. This means there is currently no data on clinical performance (phase III) of this biomarker as diagnostic tool. As calculations of sensitivity and specificity are based on a comparison between the tested method versus a standard reference method, it should be acknowledged that a readily available and reliable standard reference method is not available.

Furthermore, no studies investigate improvements of patient management when using SDMA (phase IV).

Reference ranges have been determined in studies with a decent sample size. This was done for adults and foals with foals having significantly higher normal values than adults. Reference ranges for adults have been reported in two different units: <14 microgr/dL [15,17,20], or <19 microgr/dL [18], or 0.3–0.8 micromol/L [13,14,16]. Overall, there is a low influence of extrarenal factors on SDMA in adults, with no differences between sex [15,16,17,21], age in adults [15,16,17], body weight [15,16], or body condition score [15]. Small breed differences have been noted but did not affect the cutoff values [15]. Donkeys have similar reference values as horses [17]. Foals have significantly higher values [16,20,21], ranging up to 100 microgr/dL at birth (with an upper limit of the assay of 100 microgr/dL) [20] or up to 168 microgr/dL for the first 36H [21]. SDMA concentrations decrease in the first month of life to <24 microgr/dL [20], but 2 to 6 months old foals have been reported to still have higher SDMA concentrations than adults (1.5 ± 0.4 micromol/L) [16]. Similar to SDMA in other species, factors that impact GFR also influence SDMA [12]. This has been confirmed by elevated SDMA in dehydrated horses [19] and in exercising endurance horses [13].

In conclusion, current missing evidence for the clinical application of SDMA to detect kidney disease in horses are illustrated by the fact that there are currently only few studies and limited data on kidney disease available in the literature, with no optimal cut off points and sensitivities/specificities for kidney disease available. Furthermore, the main concern is the lack of evidence for SDMA to accurately and precisely detect horses with AKI earlier than creatinine. Based on the available literature, the authors believe that SDMA may have some potential as a biomarker for kidney disease in horses (including perhaps detecting kidney disease earlier than creatinine). However, at this stage available scientific information is limited in support for SDMA used as a standard biomarker for all equine kidney disease. Nevertheless, since SDMA tests are commercially available for clinical use, and that well established reference values are available, in selected cases of suspected early or non-azotemic kidney disease and with judicious interpretation, SDMA measurements can be considered.

##### Cystatin C

In human medicine, cystatin C is a well investigated kidney biomarker that correlates better with GFR compared to serum creatinine [22]. As such, Cystatin C can offer advantages to diagnose kidney disease in situations where creatinine fails [22]. However, a recent study in dogs compared SDMA, creatinine and Cystatin C with GFR estimated by scintigraphy and found that cystatin C was inferior to both SDMA and creatinine to estimate GFR [23].

There are only 3 publications that report cystatin C measurements in horses (Table 3). Regarding the assay validation (phase I), one publication uses a human latex immunoassay, but has to conclude that the test does not work for equine samples [24]. The remaining two publications, both use a commercially available ELISA kit (MyBioSource, San Diego, CA, USA, catalogue number MBS022947) [25,26]. This kit is validated by the manufacturer for use with serum and urine samples of equine origin, but no data on this validation are available in the literature. There are currently no commercial laboratories that offer a cystatin C test for clinical purposes.

Two studies reported overlap performance data (phase II). Urine and serum cystatin C concentrations were reported to be higher in horses with AKI versus healthy horses without AKI [25]. Horses that were determined to be at risk for AKI (with colic but without endotoxemia, sepsis or systemic inflammatory response syndrome, and with absence of azotemia) were shown to have elevated serum and urine cystatin C concentrations [25]. This suggests that cystatin C has potential to detect early AKI before serum creatinine concentrations increase. In another study, serum cystatin C concentrations were significantly higher in horses infected with *Theileria equi*. This significant increase in cystatin C was noted in absence of clinically relevant azotemia and correlated to the level of parasitemia [26].

Siwinska et al. reported on clinical performance of cystatin C with the determination of optimal cut off values and their sensitivity and specificity to diagnose AKI (Phase III) [25]. The optimal cut off value for serum cystatin C was 0.53 mg/L with a sensitivity and specificity of 0.64 and 0.85; for urinary cystatin C this was at 0.20 mg/L with a sensitivity and specificity of 0.82 and 0.85 [25]. The serum cystatin C concentrations in the above mentioned *Theileria equi* infected horses reported by Ahmadpour et al. [26] were clearly higher than this identified optimal cut of value to identify AKI set by Siwinska et al. [25]. This observation also supports the idea that cystatin C is capable of detecting early AKI in absence of azotemia. Serum cystatin C was more effective to rule out AKI (so to identify healthy horses) rather than to confirm AKI, whereas urinary cystatin C was equally effective to rule in or rule out [25].

There is currently no data available on Phase IV for cystatin C in horses.

Values of serum cystatin C concentrations from healthy horses have been reported to be 0.13–0.71 mg/L [25] and 0.11 ± 0.07 ng/mL [26], and urinary concentrations 0.06–0.50 mg/L [25]. However, these values are based on relatively small numbers of horses, and currently no study has been performed to establish proper reference ranges. Furthermore, no data is available in foals.

No studies have studied how cystatin C is affected by extrarenal factors in horses, but cystatin C is reported in human studies not to be dependent on them [22].

In conclusion, current missing evidence for the clinical application of cystatin C to detect kidney disease in horses are: absence of assay validation data, absence of reference ranges, limited knowledge of extrarenal effects on cystatin C concentrations, only few studies on AKI and no studies of CKD available in the literature. Based on the available literature, the authors believe that cystatin C could hold potential as biomarker for kidney disease (including earlier detection of kidney disease compared with creatinine) and should be included in future studies. However, at this stage available scientific information is too scarce to support clinical use of this biomarker. Moreover, no tests are currently available for clinical purposes, which further limits its possibilities for clinical use at this point.

#### 3.2.2. Markers of Structural Damage: Glomerular Damage

##### Podocin

Podocin is a transmembrane signaling protein present exclusively in podocytes (highly specialized glomerular epithelial cells) and released in urine following podocyte damage [27]. Reports in humans and animals indicate that early recognition of increased urinary excretion of podocytes can facilitate the diagnosis of kidney disease [28,29]. Additionally, podocyturia is correlated with the development of glomerulopathy [30,31].

There is only 1 study that reports podocin measurements in horses (Table 4). Regarding the test validation (phase I), in the available publication podocin was detected in urine using validated qualitative method—liquid chromatography mass spectrometry in multiple reaction monitoring mode (LC-MS-MRM) and quantitatively using enzyme-linked immunosorbent assay (ELISA) [27]. For the quantitative test, the authors used a commercially available species-specific ELISA kit (MyBioSource, San Diego, CA, USA, catalogue number MBS023791). This kit is, according to the manufacturer, acceptable to use with urine samples of equine origin, but no data on this validation are available in the literature. The LC-MS-MRM method was more likely to give positive results than the ELISA [27]. There are no commercial laboratories that offer a podocin test for clinical purposes.

The same study reported overlap performance data (phase II) on 71 adult warmblood horses. With regard to the qualitative data (presence or absence), podocin was present in the urine of all horses with AKI and half of the horses at risk for AKI (with colic but without endotoxemia, sepsis or systemic inflammatory response syndrome, and with absence of azotemia) [27]. However, the presence of podocin was also detected in a few healthy horses, similar as in humans [30]. Therefore, even though presence of podocin in the urine in horses is a potential indicator of glomerular injury in subclinical and clinical kidney disease, the possibility of physiological podocyturia makes qualitative assessment of podocin clinically difficult to use. With regard to quantitative results, the urinary podocin concentration was higher in horses with AKI versus healthy horses [27]. Horses at risk for AKI had elevated podocin concentrations, but less than horses with AKI [27]. This suggests that quantitative podocin assessments have potential to confirm glomerular injury and moreover, detect it before serum creatinine concentrations increase.

The same study reported on clinical performance of the quantitative urinary podocin ELISA to diagnose AKI (Phase III). The optimal cut off value was 0.81 ng/mL with a sensitivity and specificity of 0.73 and 0.78 [27]. The podocin ELISA proved to be more effective to rule out AKI rather than to confirm AKI [27].

There is currently no Phase IV data available for podocin in horses.

The concentration of urinary podocin in healthy horses was 0.19 to 1.2 ng/mL [27]. Due to a relatively small number of examined horses, no appropriate reference ranges for podocin concentration have been established. Furthermore, no data is available in foals. The available study did not evaluate the effect of non-renal factors on podocin levels in horses. There are also no data on the effect of these factors in humans.

In conclusion, current missing evidence for the clinical application of urinary podocin to detect kidney disease in horses are: absence of assay validation data, absence of reference ranges, absence of data in foals, absence of knowledge of extrarenal effects on podocin concentrations, only 1 study with limited data available in the literature, no data on CKD. Based on the available literature, the authors believe that podocin has potential as a biomarker for kidney disease (including earlier detection of kidney disease when compared with creatinine). However, at this stage available scientific information is too scarce to support initiation of clinical use of this biomarker. Moreover, no commercial tests are currently available, which further limits its use at this point.

#### 3.2.3. Markers of Structural Damage: Tubular Damage

##### Neutrophil Gelatinase-Associated Lipocalin (NGAL)

NGAL is a small lipocalin protein synthesized by activated neutrophils and various epithelial cells, including tubular cells of the kidney. The increase in plasma and urine concentrations occurs as early as 2 h after the action of the insult [32], and is related to the degree of kidney damage. NGAL has been demonstrated in humans and small animals to help early detection of patients at risk of or with kidney disease [33,34,35].

There are two assays mentioned in equine literature (Table 5). The BioPorto porcine ELISA (Bioporto, Copenhagen, Denmark, kit 044) has been validated for NGAL measurements on equine serum. From all the different species-specific NGAL ELISA’s that are commercially available from BioPorto, only the porcine one identifies an equine NGAL signal. Equine NGAL can reliably be quantified using the porcine specific test and validation data are published [36]. Following the validation of serum samples, this commercially available porcine ELISA has also been successfully used to measure NGAL in synovial fluid [37] and peritoneal fluid [38]. The BioPorto ELISA was used for all but one publication [25]. This latter publication uses another commercially available NGAL ELISA test manufactured by MyBioSource (San Diego, CA, USA). This ELISA has been claimed to be validated by the manufacturer for use with serum and urine samples of equine origin [25], however, no details are available in the literature. The authors report similar values compared with other studies that include healthy horses and horses with kidney disease [36,38], strengthening their data. Therefore, the analytical assay validation (Phase I) has been completed for the BioPorto NGAL ELISA, but leaving some gaps for the MyBioSource test. As a side note, these tests are only available as ELISA kits with which a large number of samples has to be run at the same time making the test not applicable to a clinical scenario [37]. Currently there are no commercial laboratories offering NGAL. Human NGAL has been found to be very stable and prolonged storage or repeated freeze–thaw cycles have minimal effects on measured concentrations [37]. No data on this is available in samples from horses.

With regard to the overlap performance (Phase II), there are 2 studies reporting on kidney disease. Compared to healthy horses these studies demonstrate an increase in serum NGAL in horses with undifferentiated azotemia [36] or in serum and urine NGAL concentrations in horses with AKI [25]. Horses that were determined to be at risk for AKI (with colic but without endotoxemia, sepsis or systemic inflammatory response syndrome, and with absence of azotemia) were shown to have elevated serum and urine NGAL concentrations [25]. This suggests that NGAL has potential to detect AKI early, before serum creatinine concentrations increase. Human studies have demonstrated that NGAL can detect AKI 1–2 days earlier than traditional methods, but can also be used as a biomarker for CKD [39]. The other studies do not report on kidney disease, but focus on the aspect that NGAL also acts as an acute phase protein and increases with inflammation; this has been demonstrated in situations of equine synovial inflammation and infection [37] and equine colic related inflammation [38]. In humans, NGAL has also been suggested as a biomarker for non-renal conditions such as bacterial infections, inflammatory bowel disease, asthma, and brain and breast cancer [39].

Only one equine study reports on clinical performance with the determination of optimal cut off values and their sensitivity and specificity to diagnose AKI (Phase III) [25]. The optimal cut off value for serum NGAL was 95.2 ng/mL with a sensitivity of 0.54 and specificity of 0.93; for urinary NGAL this was at 33.1 ng/mL with a sensitivity of 0.64 and specificity of 0.71 [25]. NGAL was more effective in identifying horses without AKI rather than to confirm AKI, and did not correlate with serum creatinine concentrations [25]. These authors’ hypothesis was that the lack of correlation with serum creatinine could be related to the difference in marker pathophysiology (creatinine is a functional marker whereas NGAL is a marker of structural damage), or by the fact that NGAL also increases as a response to inflammation [25]. The fact that NGAL also increases following inflammation, similarly as in other species [34], makes its clinical use as a biomarker for kidney disease more challenging, especially since many cases with AKI also have systemic inflammation. At the same time this makes it an interesting biomarker for inflammation, in absence of kidney disease or suspicion of kidney disease. As a result, it has been suggested to have potential as a general health screening tool for horses, detecting inflammation and early kidney disease [40].

There is currently no data available on Phase IV for NGAL in horses.

Upper reference values of serum NGAL concentrations from healthy horses have been reported to range between 41.9 microg/L [38] to 103.6 microg/L [36]. The two studies with the higher values (95.20 and 103.6 microg/L) did not actively rule out inflammation in their healthy horse groups [25,36], which was done for the studies with the lower serum NGAL concentrations (41.9 and 47.7 microg/L) [38,40]. Apart from these studies that are reporting serum NGAL concentrations of small numbers of healthy horses, no study has yet truly developed reference values.

No studies have investigated whether extrarenal factors as age, sex, breed, body weight or body condition score affect NGAL concentrations. There is, however, one study in healthy Thoroughbred race horses that demonstrates that a short intense exercise has no effect on serum NGAL concentrations [40]. This cannot be extrapolated to all types of exercise, and could potentially be different for longer duration intense exercise such as endurance. As mentioned above, inflammation affects NGAL concentrations [37,38]. In other species, the urinary NGAL/creatinine ratio (UNCR) has been used [41]. No data on this ratio is currently available in horses.

There are several studies on NGAL in horses that have been presented at conferences and published as conference proceeding in internal scientific journals. Because this is not data that has been peer reviewed and did not come up in the Pubmed search as detailed in the materials and methods section, this data has not been included in the above review. However, to give the reader a brief overview of ongoing research on NGAL in horses, a short list of published conference proceedings is provided below on the topics investigated:-NGAL in neonatal foals, as a diagnostic tool to contribute to the identification of different causes of azotemia (abstract ECEIM 2020) [42].-NGAL in adult horses, overlap performance renal versus inflammation (abstract ECEIM 2018) [43].-NGAL in adult horses, following endotoxemia induction (abstract ECEIM 2018) [43].-NGAL in adult horses, as a tool to identify post-operative complications following castration (abstract ECEIM 2018) [43].-NGAL in adult horses, in bronchoalveolar fluid (abstract ECEIM 2020) [44].

In conclusion, the current missing evidence for the clinical application of NGAL to detect kidney disease in horses are: absence of established reference ranges, limited knowledge of extrarenal effects on NGAL concentrations, only few studies and limited data on kidney disease available in the literature. Based on the available literature and considering that this is a structural marker, the authors believe that NGAL has potential as an additional biomarker for kidney disease (including its possibility to detect AKI earlier than creatinine), but also potential as a marker of inflammation. However, at this stage available scientific information is still limited to support clinical use of this biomarker. Moreover, no commercial tests are currently available for clinical purposes, which further limits its possibilities for clinical use at this point.

##### Matrix Metalloproteinase 2 and 9 (MMP-2 and MMP-9)

Metalloproteinases (MMPs) are known for their ability to degrade extracellular matrix proteins. They have been demonstrated in human, and laboratory animal studies to increase in serum in relation to endotoxemia and in urine in relation to kidney damage. The latter has been suggested to be an earlier increase than creatinine [24,45,46,47,48].

There is only one study describing MMPs as kidney biomarkers in horses [24] (Table 4). In this publication by Arosalo, there is no assay validation data (phase I) described. The author refers to a previous method to measure MMPs [49], however, this is not an equine specific method, questioning the validity of the results of this study. There is currently no commercial laboratory that offers MMP measurements for clinical use.

Arosalo et al. [24] report an overlap performance study comparing colic horses undergoing exploratory laparotomy (with normal serum creatinine concentrations) to healthy horses admitted for castration; colic horses had significantly higher urinary MMP-9 complex and pro-MMP-9 activities, and higher plasma MMP-2 activity than healthy horses [24]. These authors suggest that the increase in plasma MMP-2 activity is related to endotoxemia, and that the increase in urinary MMP-9 complex and pro-MMP-9 activities is indicative of AKI [24]. Therefore, urinary MMP activity can also in horses be considered an early marker of AKI with the potential to identify AKI cases before serum creatinine concentrations increase.

There is no clinical performance data (phase III), usefulness data (phase IV), reference ranges or information on extra-renal factors available in the literature on MMPs as kidney biomarkers in horses.

In conclusion, current missing evidence for the clinical application of MMP-9 to detect kidney disease in horses are: absence of assay validation data, absence of reference ranges, limited knowledge of extrarenal effects, only 1 study on potential early AKI but no confirmed AKI cases available in the literature. The authors believe that the current available literature on MMP-9 does not allow to make statements about its potential as a biomarker for kidney disease. At this stage, no scientific information is available to support clinical use of this biomarker. Moreover, no commercial tests are currently available for clinical purposes.

##### *N*-Acetyl-beta-D-glucosaminidase (NAG)

NAG is a large lysosomal protein of the proximal tubules and is not reabsorbed nor filtered in the glomeruli due to its size [50]. As such, measuring NAG in urine has been hypothesized to be useful for detection of kidney disease. Because the proximal tubules are a key part of AKI pathology and the acute-on-chronic disease cycles that often occur during CKD, NAG is theoretically well suited as a biomarker of kidney disease.

NAG has been evaluated in one study in horses, where a photometric assay was examined on equine urine using a biochemical analyser [50] (Table 4). Phase 1 results are available from the study on 7 non-azotemic and 7 azotemic horses. The intra-run coefficient of variation for Urine NAG/Urine Creatinine was 20% whereas the intrarun coefficient of variation was 3.2% with a high linearity during serial dilutions. NAG concentrations have been reported to be affected by urinary pH, with alkaline urine leading to a drop in concentrations; this may be applicable in the interpretation in equine patients, given the alkalinity of equine urine [50]. However, further work is needed to ascertain the effect of this on different assays [51].

Despite the small sample size, NAG concentrations are significantly higher in azotemic horses compared to non-azotemic horses [50]. However, the number of horses enrolled in the study is inadequate to assess performance in health and disease (phase II).

There are no clinical performance data (phase III), usefulness data (phase IV) or reference ranges for horses available in the literature. Due to the limited number of horses in the study, it is not possible to establish reference ranges or diagnostic accuracy.

In conclusion, due to the lack of evidence, there is no support for the clinical use of NAG as a biomarker of kidney disease in horses. NAG should be included in futures studies of AKI and CKD in horses as it may have potential. As NAG can be run on a commercial analyser it could easily be included in screening studies in the future.

**Table 2 animals-12-02678-t002:** List of studies on symmetric dimethylarginine (SDMA) concentrations in blood in horses and its analytical and clinical characteristics.

	Study	Assay Used	Nr. of Horses Included	Nr. of Horses with AKI or CKD	Comparison to Traditional Kidney Assessment	Association with Clinical State	Validation Phase	Reference Range Healthy Horses	Extrarenal Factors Influencing Reference Range	Limitations
1	Lo, H.C., Winter, J.C., Merle, R., Gehlen, H. Symmetric dimethylarginine and renal function analysis in horses with dehydration. Equine Vet J. 10 June 2021 [19].	DLD SDMA ELISA kit (DLD Diagnostika GmbH) (validated by [16])	41 horses with dehydration	1 with AKI for 48 h, 1 with AKI for 12 h	Good correlation with creatinine, not with urea	Dehydration severity AKI: SDMA elevated 2 cases with normal creatinine but with elevated SDMA (subclinical AKI?) Lower in survivors than non-survivors, but not statistical	Phase II	n/a	Dehydration	Limited AKI cases
2	Bozorgmanesh, R., Thornton, J., Snyder, J., Fletcher, C., Mack, R., Coyne, M., Murphy, R., Hegarty, E., Slovis, N. Symmetric dimethylarginine concentrations in healthy neonatal foals and mares. J Vet Intern Med. 2021 Nov;35(6):2891–2896 [20].	IDEXX SDMA high throughput immunoassay (validated by [15])	104 healthy periparturient TB mares (last month of gestation) and 125 their full-term foals (first month of life)	0	SDMA higher than adult values during first month; creatinine only for the first 12 h after birth	n/a	n/a	Birth 0–100 microgr/dL; 1–4 days old 0–85 microgr/dL; 5–10 days old 0–36 microgr/dL; 20–30 days old 0–24 microgr/dL; mares <14 microgr/dL	No correlation between mare SDMA and foal SDMA	Upper limit assay 100 microgr/dL
3	Gough, R.L., McGovern, K.F. Serum symmetric dimethylarginine concentration in healthy neonatal Thoroughbred foals. Equine Vet J. 2021 [21].	IDEXX ELISA (presumably the same as validated [15])	120 healthy neonatal TB foals <36 h	0	Correlation with creatinine and urea	n/a	n/a	<168 microgr/dL	Correlation with age, no difference between sexes	No consideration of subclinical disease
4	Ertelt, A., Stumpff, F., Merle, R., Kuban, S., Bollinger, L., Liertz, S., Gehlen, H. Asymmetric dimethylarginine-A potential cardiac biomarker in horses. J Vet Cardiol. 2021 Feb;33:43–51 [14].	Commercial human ELISA & cross checked with liquid chromatograph triple quadrupole mass spectrometry	78 adult horses with unknown medical history, 23 adult horses with confirmed structural cardiac disease	0	n/a	No difference between horse with or without cardiac disease	(Phase I) Phase II	0.3–0.8 micromol/L	n/a	No information on kidney status
5	Schott, H.C., 2^nd^, Gallant, L.R., Coyne, M., Murphy, R., Cross, J., Strong-Townsend, M., Szlosek, D., Yerramilli, M., Li, J. Symmetric dimethylarginine and creatinine concentrations in serum of healthy draft horses. J Vet Intern Med. 2021 Mar;35(2):1147–1154 [15].	Immunoassay IDEXX, added to an automated multichannel chemistry analyzer allowing high throughput, compared to liquid chromatography-mass spectroscopy	165 healthy adult (>0.5 years old) draft horses	1 AKI (or dehydration), 7 elevated SDMA with normal—mild increased creatinine	Correlated to creatinine, not to urea nitrogen	n/a	Phase I	Draft horses <14 microgr/dL	Age: no correlation Sex: no difference Breed: lower in Percherons and Belgians than in Percherons Weight or BCS: no correlations	
6	Siwinska, N., Zak, A., Paslawska, U. Detecting acute kidney injury in horses by measuring the concentration of symmetric dimethylarginine in serum. Acta Vet Scand. 2021 Jan 15;63(1):3 [18].	Enzyme immunoassay (EIA) on the Beckmann Coulter analyzer (IDEXX) (mentioned to be validated by [17])	30 healthy adult horses, 30 horses in risk group (with gastro-intestinal disease and/or receiving PBZ/gentamicin), 11 horses with AKI	11 AKI	Correlated to creatinine and urea, SG and urinary protein.	Health category: SDMA comparable between healthy and at risk, but increased in AKI. At risk group showed no increase in SDMA but increase in traditional renal values (urinary protein and urinary protein/cr ratio and urinary GGT/creatinine ratio) -> limited use of SDMA in subclinical cases	Phase II	<19 microgr/dL	n/a	
7	Siwinska, N., Zak, A., Slowikowska, M., Niedzwiedz, A., Paslawska, U. Serum symmetric dimethylarginine concentration in healthy horses and horses with acute kidney injury. BMC Vet Res. 2020 Oct 20;16(1):396 [16].	Commercially available and non-species specific enzyme imunoassay ELISA kit (DLD Diagnostika GmbH) from Labor der SYNLAB vet GmHb (Ausburg, Germnay)—validation info in supplementary data)	53 healthy horses (17 2–6 month old foals and 36 adults) and 23 horses with AKI	23 AKI	Correlated with creatinine	AKI: higher than healthy	Phase I (validation in supplementary data) Phase II	Healthy foals 1.5 ± 0.4 micromol/L healthy adults 0.53 ± 0.14 micromol/L	Age: higher in foals, no effect in adults Sex: no effect Body weight: no effect	
8	Ertelt, A., Merle, R., Stumpff, F., Bollinger, L., Liertz, S., Weber, C., Gehlen, H. Evaluation of Different Blood Parameters From Endurance Horses Competing at 160 km. J Equine Vet Sci. 2021 [13].	Commercially available fast ELISA run by DLD Gesellschaft fur Diagnostika und medizinische Gerate mbH, Hamburg, Germany—validated	52 healthy endurance horses—finishers versus non-finishers	0	n/a	Pre vs. post-race	Phase I—validation info provided Phase II	Before racing all in ref range of 0.3–0.8 micromol/L	Exercise and finishing endurance: SDMA higher after the race in finishers (7 above ref range) Distance: more increased SDMA with longer distance	
9	Frączkowska, K., Trzebuniak, Z., Żak, A., Siwińska, N. Measurement of Selected Renal Biochemical Parameters in Healthy Adult Donkeys Considering the Influence of Gender, Age and Blood Freezing. Animals. 2021 Jun 11;11(6):1748 [17].	Validated enzyme immunoassay performed on a Beckman Coulter analyzer (USA)	65 healthy adult donkeys	0	n/a	n/a	No analytical validation data available in study, but mentioned to be validated	<14 microgr/dL	Freezing and storing for 1 year: no effect Sex: no effect Age: no effect	

Legend: AKI—acute kidney injury; CKD—chronic kidney disease; SDMA—symmetric dimethylarginine; ELISA—enzyme linked immunosorbent assay; SG—specific gravity. References: [13,14,15,16,17,18,19,20,21].

**Table 3 animals-12-02678-t003:** List of studies on cystatin C and their analytical and clinical characteristics.

	Study	Assay Used	Sample Type	Nr. of Horses Included	Nr. of Horses with AKI or CKD	Comparison to Traditional Kidney Assessment	Association with Clinical State	Validation Phase	Reference Range Healthy Horses	Extrarenal Factors Influencing Reference Range	Limitations
1	Siwińska, N., Żak, A., Pasławska, U. Evaluation of Serum and Urine Neutrophil Gelatinase-associated Lipocalin and Cystatin C as Biomarkers of Acute Kidney Injury in Horses. J Vet Res. 2021 May 16;65(2):245–252 [25].	Commercially available ELISA kits (MyBioSource, San Diego, CA, USA, catalogue number MBS022947) validated by the manufacturer for use with serum and urine samples of equine origin	Serum (S), urine (U)	30 healthy horses 30 horses at risk of AKI (colics or nephrotoxic drugs) 11 horses with AKI	30 at risk of AKI 11 with AKI	S cystatin C correlated to urea U cystatin C correlated to urea, creatinine, urine protein and urine GGT/protein ratio	U & S cystatin C higher in colics and AKI horses vs. non AKI	Phase I: mentioned that test is validated but no validation data available Phase II overlap healthy—AKI risk—AKI Phase III—Se & Sp best cut off value S Cystatin C 0.53 mg/L with Se 0.64, Sp 0.85, PPV 0.44, NPV 0.94; best cut off value U Cystatin C 0.20 mg/L with Se 0.82, Sp 0.85, PPV 0.50, NPV 0.96 More effective detecting negative than positive results	S median 0.25 mg/L (1st and 3rd quartile: 0.19–0.37 mg/L) U median 0.1 mg/L (1st and 3rd quartile 0.07–0.13 mg/L)	n/a	Small nr. of cases
2	Ahmadpour, S., Esmaeilnejad, B., Dalir-Naghadeh, B., Asri-Rezaei, S. Alterations of cardiac and renal biomarkers in horses naturally infected with Theileria equi. Comp Immunol Microbiol Infect Dis. 2020 May 30;71:101502 [26].	Commercially available ELISA kits (MyBioSource, San Diego, CA, USA, catalogue number MBS022947)	serum	28 horses (3–4 year old) infected with Theileria equi; 20 healthy control horses	0	Correlation to urea and creatinine; minimal increase in creatinine and urea, but significant increase in cystatin C -> can identify kidney damage earlier than creatinine and urea	Cystatin C significantly increased in horses with Theleria equi infection (> cut off value set by Siwinska et al. 2021); association to the level of parasitemia.	Phase II overlap healthy—Theileria equi	0.11 ± 0.07 ng/mL	n/a	Small nr. of horses
3	Arosalo, B.M., Raekallio, M., Rajamäki, M., Holopainen, E., Kastevaara, T., Salonen, H., Sankari, S. Detecting early kidney damage in horses with colic by measuring matrix metalloproteinase -9 and -2, other enzymes, urinary glucose and total proteins. Acta Vet Scand. 2007 Jan 23;49(1):4 [24].	Dako Cytomation latex immunoassay^®^ (Dako Cytomation, Glostrup, Denmark) based on turbidimetry (particle-enhanced tur- bidimetric immunoassay, PETIA) using human cystatin- C-specific antibodies	serum	7 healthy control horses admitted for elective castration; 5 sick horses admitted for colic and undergoing emergency celiotomy	0 (all colic horses had serum creatinine below 162 micromol/L)	n/a	No correlation noted	n/a	Cystatin-C concentrations were below the threshold level of the test (<0.39 mg/L)	n/a	Human cystatin-C test did not work, indicating no or very little cross-reactivity between human and equine cystatin-C specific antibody

Legend: AKI—acute kidney injury; CKD—chronic kidney disease; ELISA—enzyme linked immunosorbent assay; Se—sensitivity; Sp—specificity; PPV—positive predictive value; NPV—negative predictive value; S—serum; U—urine. References [24,25,26].

**Table 4 animals-12-02678-t004:** List of studies on other biomarkers on kidney disease and their analytical and clinical characteristics.

	Study	Assay Used	Sample Type Used	Nr. of Horses Included	Nr. of Horses with AKI or CKD	Comparison to Traditional Kidney Assessment	Association with Clinical State	Validation Phase	Reference Range Healthy Horses	Extrarenal Factors	Limitations
** *Podocin* **											
1	Siwińska, N., Pasławska, U., Bąchor, R., Szczepankiewicz, B., Żak, A., Grocholska, P., Szewczuk, Z. Evaluation of podocin in urine in horses using qualitative and quantitative methods. PLoS One. 2020 Oct 15;15(10):e0240586 [27].	Qualitative method—liquid chromatography mass spectrometry in multiple reaction monitoring mode (LC-MS-MRM) Quantitative method—commercially available species-specific ELISA kit (MyBioSource, San Diego, CA, USA)	Urine	30 healthy horses 30 horses at risk of AKI (colics or nephrotoxic drugs) 11 horses with AKI	30 at risk of AKI 11 with AKI	Higher serum creatinine and urea levels, as well as proteinuria, increased UPC, GGT: Crea ratio, and FENa were associated with the presence of podocin in the urine sample; low positive correlation between the podocin level and UPC in all the examined horses, as well as between podocin and FENa	Podocin higher in horses with AKI vs. healthy horses; horses at risk for AKI have elevated podocin, but less than horses with AKI	Phase I: validated qualitative method; mentioned that quantitative test is validated but no validation data mentioned Phase II overlap healthy—AKI risk—AKI Phase III—Se & Sp best cut off value 0.81 ng/mL with Se 0.73, Sp 0.78, PPV 0.38, NPV 0.94; More effective detecting negative than positive results	0.19–1.2 ng/mL	n/a	Small nr. of cases
** *Metalloproteinases (MMP)* **											
1	Arosalo, B.M., Raekallio, M., Rajamäki, M., Holopainen, E., Kastevaara, T., Salonen, H., Sankari, S. Detecting early kidney damage in horses with colic by measuring matrix metalloproteinase -9 and -2, other enzymes, urinary glucose and total proteins. Acta Vet Scand. 2007 Jan 23;49(1):4 [24].	SPS-PAGE gelatin zymography. Zymograms were analyzed for total gelatinolytic activity, complex-, proMMP-9, active MMP-9, and MMP-2 forms. The assays were performed as previously described	Plasma and urine	7 healthy control horses admitted for elective castration; 5 sick horses admitted for colic and undergoing emergency celiotomy	0 (all colic horses had serum creatinine below 162 micromol/L)	Creatinine higher in colic group but within reference ranges (no azotemia)	Plasma MMP-2 was higher in colic horses than controls—suggested to be related to endotoxemia Urinary MMP-9 complex and proMMP-9 were higher in colic horses than controls—suggested to be related to AKI	No exact values reported in the publication (only boxplot graphs)	n/a	n/a	Small nr. of cases
** *N-acetyl-β-D-glucosaminidase (NAG)* **											
1	Bayless, R.L., Moore, A.R., Hassel, D.M., Byer, B.J., Landolt, G.A., Nout-Lomas YS. Equine urinary *N*-acetyl-β-D-glucosaminidase assay validation and correlation with other markers of kidney injury. J Vet Diagn Invest. 2019 Sep;31(5):688–695 [50].	Photometric assay (*N*-acetyl-β-D-glucosaminidase; Roche Diagnostics, Mannheim, Germany)	Urine	7 hospitalized horses with azotemia and 7 non-azotemic horses with normal creatinine	0	Urine NAG was significantly correlated with urinary fractional excretion of sodium and plasma creatinine.	NAG was higher in azotemic horses compared to non-azotemic horses	Phase 1: No reference standard to compare with validation data acceptable. Phase II: Partially done with increase during azotemia	n/a	NAG concentration affected by urine pH	Small nr. of patients

Legend: AKI—acute kidney injury; CKD—chronic kidney disease; FENa—fractional excretion of sodium; Se—sensitivity; Sp—specificity; PPV—positive predictive value; NPV—negative predictive value; ELISA—enzyme linked immunosorbent assay. References [24,27,50].

**Table 5 animals-12-02678-t005:** List of studies on neutrophil gelatinase-associated lipocalin (NGAL) in horses and its analytical and clinical characteristics.

	Study	Assay Used	Sample Type	Nr. of Horses Included	Nr. of Horses with AKI or CKD	Comparison to Traditional Kidney Assessment	Association with Clinical State	Validation Phase	Reference Range Healthy Horses	Extrarenal Factors Influencing Reference Range	Limitations
1	Siwińska, N., Żak, A., Pasławska, U. Evaluation of Serum and Urine Neutrophil Gelatinase-associated Lipocalin and Cystatin C as Biomarkers of Acute Kidney Injury in Horses. J Vet Res. 2021 May 16;65(2):245–252 [25].	Commercially available ELISA kits (MyBioSource, San Diego, CA, USA, catalogue number MBS087573) validated by the manufacturer for use with serum and urine samples of equine origin.	Serum (S), urine (U)	30 healthy horses 30 horses at risk of AKI (colics, or nephrotoxic drugs) 11 horses with AKI	30 at risk of AKI 11 with AKI	S NGAL low correlations with BUN and urine GGT/creatinine ratio. Does not correlate with creatinine U NGAL good correlations with urine protein, urine GGT/creatinine ratio and FENa. No correlation with creatinine and BUN	U & S NGAL higher in colics and AKI horses vs. non AKI	Phase I: mentioned that test is validated but no validation data available Phase II overlap healthy—AKI risk—AKI Phase III—Se & Sp best cut off value S NGAL 95.2 ng/mL with Se 0.54, Sp 0.93, PPV 0.60, NPV 0.92; best cut off value U NGAL 33.1 ng/mL with Se 0.64, Sp 0.71, PPV 0.30, NPV 0.90 More effective detecting negative than positive results	S median 50.5 ng/mL (25–95.20 ng/mL) U median 20.7 ng/mL (8.5–41.7 ng/mL)	n/a	No assessment of inflammation
2	Frydendal, C., Nielsen, K.B., Berg, L.C., van Galen, G., Adler, D.M.T., Andreassen, S.M., Jacobsen, S. Influence of clinical and experimental intra-articular inflammation on neutrophil gelatinase-associated lipocalin concentrations in horses. Vet Surg. 2021 Apr;50(3):641–649 [37].	Bioporto porcine NGAL ELISA (kit 044), Copenhagen, Denmark	Serum (S), synovial fluid (SF)	19 healthy horses 7 horses with experimental LPS synovial inflammation 12 horses with experimental synovial inflammation by local analgesics 20 clinical cases with septic synovitis	0	n/a	Synovial inflammation: NGAL increases I S and in SF (SF > S) NGAL: SF sepsis > SF inflammation > healthy	Phase II, however not including kidney disease Timing of increase after insult: 12–24 h in S, 6–12 h in SF	n/a	n/a	No assessment of kidney disease in these patients
3	Winther, M.F., Haugaard, S.L., Pihl, T.H., Jacobsen, S. Concentrations of neutrophil gelatinase-associated lipocalin are increased in serum and peritoneal fluid from horses with inflammatory abdominal disease and non-strangulating intestinal infarctions. Equine Vet J. 2022 May 31 [38].	Bioporto porcine NGAL ELISA (kit 044), Copenhagen, Denmark	Serum (S), peritoneal fluid (PF)	270 horses with acute colic 9 healthy horses	0 (Kidney disease was excluded except for horses with marginally elevated creatinine that returned to normal after fluids)	n/a	Simple obstructions and strangulating obstructions similar S and PF NGAL as healthy (low). NGAL increased in non-strangulating intestinal infarction and inflammatory abdominal disease in NGAL higher in horses with longer duration of colic	Phase II, however not including kidney disease	S: median 21.0, range 4.4–41.9 microg/L PF: median 9.5, range 5.0–239.1 microg/L	n/a	
4	Jacobsen, S., Berg, L.C., Tvermose, E., Laurberg, M.B., van Galen, G. Validation of an ELISA for detection of neutrophil gelatinase-associated lipocalin (NGAL) in equine serum. Vet Clin Pathol. 2018 Dec;47(4):603–607 [36].	Bioporto porcine NGAL ELISA (kit 044), Copenhagen, Denmark	serum	20 horses with elevated creatinine >130 micromol/L, 19 healthy horses with creatinine <100 micromol/L	20 kidney cases	Yes, elevated creatinine associated with elevated NGAL	NGAL differs between horses with normal creatinine versus increased creatinine	Full phase I assay validation. Overlap performance phase II kidney	7.4–103.6 microgr/L, median 21.5 microgr/L, mean 28.5 microgr/L	n/a	130 micromol/L creatinine cut off is relatively low to identify AKI, no assessment of inflammation
5	Marnie Flick, Anne Mette Lindberg Vinther, Stine Jacobsen, Lise C. Berg, Marina Gimeno, Denis Verwilghen, Wade Howden, Kate Averay, Gaby van Galen (2021). Is serum Neutrophil gelatinase-associated lipocalin (NGAL) concentration in horses affected by racing? A pilot study. JVCP [40]	Bioporto porcine NGAL ELISA (kit 044), Copenhagen, Denmark	serum	14 healthy TB racehorses: pre-racing and 24 H post-racing	0	Yes compared to creatinine and SAA. All normal creatinine and normal SAA	n/a	n/a	16.98–47.70 ng/mL	Racing has no effect: pre- and post-racing similar values of NGAL	No other disciplines than racing

Legend: AKI—acute kidney injury; CKD—chronic kidney disease; NGAL—neutrophil gelatinase-associated lipocalin; ELISA—enzyme linked immunosorbent assay; Se—sensitivity; Sp—specificity; PPV—positive predictive value; NPV—negative predictive value; S—serum; U—urine; SF—synovial fluid; PF—peritoneal fluid. References [25,36,37,38,40].

In Table 6 an overview is given off the molecular and physiological information of all biomarkers discussed above.

## 4. Discussion

This systematic review of biomarkers of kidney disease in the horse demonstrates that although some data are available, overall, there is a lack of scientific data and a lack of strong evidence for the clinical use of these biomarkers in the approach to horses with kidney disease. SDMA is currently the best studied and the only one for which a commercially available test is available for sample submission to a commercial lab. Despite this, the evidence to suggest clinical use of SDMA is not strong, mainly because of the lack of evidence that SDMA can detect kidney disease earlier than creatinine can. There is currently some but limited evidence on NGAL and cystatin C. However, these studies show strong potential for future use for both these biomarkers to diagnose kidney disease, including earlier detection of kidney disease than creatinine. The data on other biomarkers described in the literature is very limited and based on a single paper, not allowing for conclusions.

Limitations of the evidence included in the review are mainly reflected by the small number of studies investigating these biomarkers, the small amount of studies that include actual kidney cases, and the incomplete validation process of these biomarkers. Limitations of the review processes used in this systematic review include the fact that some relevant studies could have been missed, and that the conclusion on evidence is based not only on the literature but also on the opinion of the authors. The authors have tried to remain as objective as possible.

This systematic review could have implications for practice, policy, and future research. It clearly demonstrates where the current gaps are and what future research efforts should be directed at. Initial reports of new biomarkers often provide exciting clues to the pathophysiology of diseases, improved diagnostic capabilities, and perhaps, even better care for our equine patients. However, moving from research to clinical application demands for replication of results in multiple settings, clear experimental evidence supporting their clinical diagnostic role, and ideally, trials demonstrating that using the biomarker improves our practice. Some authors even suggest that despite the huge efforts made, no novel kidney biomarkers have added clear-cut additional value on clinical decision making in human medicine above what is already provided by established cheap and easily available diagnostic tools such as proteinuria and creatinine concentrations (with the exception of anti-phospholipase A2 receptor antibodies to diagnose membranous nephropathy) [59,60]. It has been suggested that until these steps are taken, the use of new biomarkers in human medicine should be reserved for investigative purposes, and is suggested to be restrictive; their diagnostic use is limited to identified high-risk patients with established risk factors who would benefit from aggressive management [61].

Few biomarkers are tested beyond Phase I and even if they make it to Phase II and Phase III, the numbers are small, and the diagnostic accuracy rarely evaluated. In a single referral hospital setting it is time consuming and expensive to obtain adequate sample numbers. Multicenter collaborations or multicenter biobanks with urine and serum samples for larger scale testing on horses with kidney disease and the availability of data from previous tests on these samples could be a solution.

Standard clinical use of the discussed biomarkers in equine medicine is at this stage not recommended, or simply impossible because of lack of testing possibilities that are offered by commercial laboratories. Similar as in human medicine, availability of a test, low cost, ease of performance and obtaining results rapidly enough to influence clinical decision making are desirable characteristics [60,61], and determine eventually if equine clinicians will or can use the biomarkers. Nevertheless, in the future, these or other kidney biomarkers perhaps become a valuable part of clinical diagnostics, and it is worthwhile discussing how they can be integrated in a clinical approach of a kidney diseased horse. As a first suggestion, they could be valuable assets to detect kidney disease especially in non-azotemic horses. As such they could be included in regular health screenings or screenings of horses that have a high risk for kidney disease (for example critically ill horses, or prior to or during administration of nephrotoxic drugs). When horses are already azotemic, measuring additional biomarkers to detect kidney disease has limited value, except when localization of the lesions contributes to improved patient management (glomerular versus tubular). Furthermore, creation of a serum and urinary biomarker panel to assist in diagnosing kidney disease could be of benefit. Although the measurement of creatinine as an indicator of kidney function has limitations as discussed in the introduction, it still fulfils requirements for use in everyday clinical practice. It can be easily, reliably and inexpensively measured with a good sensitivity and specificity [54]. Therefore, creatinine is unlikely to disappear anytime soon from our diagnostic toolbox.

## 5. Conclusions

In conclusion, this systematic literature review on biomarkers of equine kidney disease shows that there are multiple biomarkers with the potential to diagnose kidney problems. However, at the moment there are only few studies available and clearly more data is needed before these biomarkers can be applied and recommended in our daily practice.

## Figures and Tables

**Figure 1 animals-12-02678-f001:**
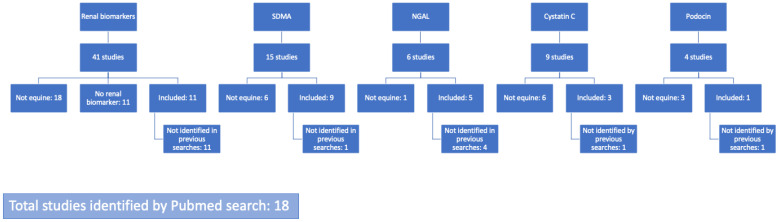
Flow diagram of the literature search for studies on biomarkers on equine kidney disease. SDMA—symmetric dimethylarginine; NGAL—neutrophil gelatinase-associated lipocalin. Searches were always in combination with the search word horse or equine.

**Table 1 animals-12-02678-t001:** List of the variables described for each biomarker for kidney disease in horses.

Type of Information	Details	Data Obtained from
Molecular information	Type of molecule Size	Other studies, no systematic literature search
Physiological information	Origin Function in the body Kidney physiology Structural or functional marker	Other studies, no systematic literature search
Analytical information	Which assays are available (type, brand, developed for which species, commercially available?) Analytical validation of assays for horses (phase I validation Samples to be used: blood or urine Equine reference ranges Extrarenal factors affecting reference ranges	Systematic literature search as specified in Materials and Method section
Clinical information	Validation data (overlap/clinical/outcome; phase II, II IV) Comparison to other biomarkers Results of individual studies Results of combined studies	Systematic literature search as specified in Materials and Method section
Author conclusions	What knowledge or development is still required for clinical use in horses? Level of evidence for or against use in horses	Discussion between authors, based on the above literature data

No statistical analyses were performed for this systematic review.

**Table 6 animals-12-02678-t006:** Overview of molecular and physiological information on the identified kidney biomarkers.

	Molecular Information		Physiological Information				References
	Type of Molecule	Size	Origin	Function in the Body	Kidney Physiology	Extra Renal Factors Influencing Biomarker in Other Species	
** *Functional biomarkers* **							
SDMA	Methylated aminoacid	202.25	Generated intracellularly during protein degradation as analogues of L-arginine; enantiomer of ADMA	Methylation of arginine residues appears to be required for RNA processing, protein shuttling and signal transduction. SDMA has an indirect effect on NO synthesis by interfering with tubular arginine absorption	Almost all SDMA is excreted by kidneys, concentration proportional to GFR. Some hepatic clearance also occurs	Factors that impact GFR	[12,52,53,54]
Cystatin C	Cationic and non-glycosylated protein	13 kDa	Produced by all nucleated cells of the body and is released into the intravascular compartment at a constant rate	Protease inhibitor	Filtered from the blood through the glomerulus and catabolized, but not secreted, by the proximal tubular cells. Concentration proportional to GFR.	Inflammation, metabolic diseases, and even the interval between feeding and serum collection. Factors that impact GFR. Independent from age, sex, race, muscle mass	[22,55]
** *Markers of structural damage* **							
***Glomerular damage***							
Podocin	Podocyte associated molecule	450 kDa	Present exclusively in podocytes (specialized glomerular epithelial cells), especially near the slit diaphragms	Podocin is one of the proteins that constitute the slit-diaphragms of the glomerulus (modified tight junction that facilitates size and charge dependent permselectivity)	Released in urine following podocyte damage	Physiological podocyturia, more in concentrated urine	[27,56]
***Tubular damage***							
NGAL	Lipocalin protein	25 kDa	Activated neutrophils, and various epithelial cells, kidney tubular cells	Plays a role in iron metabolism and innate immunity to bacteria, kidney development, growth factor for stimulated epithelia (inflammation, bacterial infection or neoplasia), tubular regeneration after injury, protective effect for kidney ischaemia-reperfusion injury	Highly expressed in response to tubular injury	Inflammation	[39,57]
MMP	Metalloproteinases	Pro-MMP 72 kDa; MMP-2 59–62 kDa; proMMP-9 68–92 kDa (dependent on activation status)	All cells	Degradation of extracellular matrix proteins, role in apoptosis	Small molecules pass through glomerular filtration, but in most healthy animals are reabsorbed in the proximal tubules. Excretion in urine increases with tubular or glomerular damage	Endotoxemia influences plasma levels	[24]
NAG	Enzyme	130–140 kDa	Lysosomal enzyme in proximal tubular epithelium	Lysosomal enzyme	Released in case of tubular damage	Physiologic exostosis into the tubular lumen. Urine concentration -> NAG: creatinine ratio Instability in alkaline urine	[50,58]

Legend: SDMA—symmetric dimethylarginine; NGAL—neutrophilgelatinase-associated lipocalin; MMP—matrix metalloproteinases; NAG—*N*-acetyl-β-D-glucosaminidase; GFR—glomerular filtration rate.

## Data Availability

Not applicable.

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
