# Peer review of "Biomarkers of Kidney Disease in Horses: A Review of the Current Literature"

_animals, 2022, doi:10.3390/ani12192678_

Round 1
Reviewer 1 Report
The authors report a systematic review on biomarkers of kidney disease. The paper is well structured and well written and clear. It is of high interest and could be suitable for publication after minor revision.
Simple summary:
Line 12: it can be …. The sentence is not clear, please erase and start directly with … the one or please rephrase it.
Mat and met
Line 237: please erase the comma and add a space.
Line 237-239: please better explain the sentence. Not easy to read.
Line 241: please erase “of horses”
Line 247: please change as follow: … .. to creatinine ratio (UP/UC)
Line 500: the author, not the reader
Line 506: for castration (please erase the “a”)
Author Response
20th of September 2022
Dear reviewers and editor,
We appreciate that you have taken the time to review our manuscript, and have taken all your comments into account. All our replies to your comments are listed below and changes in the manuscript are marked with track changes.
Yours sincerely
Gaby van Galen, as representative for all authors
Reviewer 1
The authors report a systematic review on biomarkers of kidney disease. The paper is well structured and well written and clear. It is of high interest and could be suitable for publication after minor revision.
Simple summary:
Line 12: it can be …. The sentence is not clear, please erase and start directly with … the one or please rephrase it.
Author’s reply: this has been rephrased
Mat and met
Line 237: please erase the comma and add a space.
Author’s reply: this has been adjusted
Line 237-239: please better explain the sentence. Not easy to read.
Author’s reply: this has been rephrased
Line 241: please erase “of horses”
Author’s reply: this has been erased
Line 247: please change as follow: … .. to creatinine ratio (UP/UC)
Author’s reply: this has been changed
Line 500: the author, not the reader
Author’s reply: this has been adapted
Line 506: for castration (please erase the “a”)
Author’s reply: this has been erased

Reviewer 2 Report
The review presents new equine kidney biomarkers based on the published literature. The authors stressed that available data on promising new markers are not enough and theses markers should be better investigated before application in clinical practice. The topic is suitable for the special issue. The manuscript can be accepted for publication.
The authors properly defined important terms at the beginning of the manuscript. The inclusion and exclusion criteria are well described and the applied approach is clear. The style of the presentation of the information is very helpful for understanding of the message.
Lines 99-102: Please, refine the following sentence “AKI and CKD are also possible without azotemia, for example in unilateral disease where the second kidney can compensate for the loss of function of one kidney, or damage that is impacting a small amount of nephrons.” How these conditions can be diagnosed? It is not well clarified.
Lines 149-150: “How well does the test function from an analytical point of view, ie intra- and inter-assay variations, linearity under dilution.” – this sentence is not complete, many other factors can influence the accuracy, precision, specificity and other characteristics of the analytical method, therefore the authors can add “and other factors”.
Tables 2 - 6 are very informative but it will be better if they are presented in landscape view.
Author Response
20th of September 2022
Dear reviewers and editor,
We appreciate that you have taken the time to review our manuscript, and have taken all your comments into account. All our replies to your comments are listed below and changes in the manuscript are marked with track changes.
Yours sincerely
Gaby van Galen, as representative for all authors
Reviewer 2:
The review presents new equine kidney biomarkers based on the published literature. The authors stressed that available data on promising new markers are not enough and theses markers should be better investigated before application in clinical practice. The topic is suitable for the special issue. The manuscript can be accepted for publication.
The authors properly defined important terms at the beginning of the manuscript. The inclusion and exclusion criteria are well described and the applied approach is clear. The style of the presentation of the information is very helpful for understanding of the message.
Lines 99-102: Please, refine the following sentence “AKI and CKD are also possible without azotemia, for example in unilateral disease where the second kidney can compensate for the loss of function of one kidney, or damage that is impacting a small amount of nephrons.” How these conditions can be diagnosed? It is not well clarified.
Author’s reply: this sentence has been re-written, and hopefully it is clear now what is meant.
Lines 149-150: “How well does the test function from an analytical point of view, ie intra- and inter-assay variations, linearity under dilution.” – this sentence is not complete, many other factors can influence the accuracy, precision, specificity and other characteristics of the analytical method, therefore the authors can add “and other factors”.
Author’s reply: this has been added
Tables 2 - 6 are very informative but it will be better if they are presented in landscape view.
Author’s reply: we fully agree, and this was also submitted in landscape view. However with the uploading of the document it was not possible to get the tables in landscape, and they unfortunately automatically changed to portrait view. We are confident that the editors will shape this in a nice format so that readability is enhanced.
